# Characterization and Mechanism of Tea Polyphenols Inhibiting Biogenic Amine Accumulation in Marinated Spanish Mackerel

**DOI:** 10.3390/foods12122347

**Published:** 2023-06-12

**Authors:** Zhe Xu, Jiale Chang, Jiamin Zhou, Yixin Shi, Hui Chen, Lingyu Han, Maolin Tu, Tingting Li

**Affiliations:** 1Key Laboratory of Biotechnology and Bioresources Utilization, College of Life Sciences, Dalian Minzu University, Dalian 116029, China; dlpuxz@163.com (Z.X.); cjl15735751463@163.com (J.C.); haoshaoyu321@163.com (J.Z.); hanlingyu1001@126.com (L.H.); 2School of Food Science and Technology, National Engineering Research Center of Seafood, Collaborative Innovation Center of Seafood Deep Processing, Dalian Polytechnic University, Dalian 116034, China; yixshi@outlook.com; 3Key Laboratory of Animal Protein Food Deep Processing Technology of Zhejiang Province, College of Food and Pharmaceutical Sciences, Ningbo University, Ningbo 315832, China; realcrital@126.com; 4Key Laboratory of Marine Fishery Resources Exploitment & Utilization of Zhejiang Province, Zhejiang University of Technology, Hangzhou 310014, China; tumaolin012@163.com

**Keywords:** putrescine, Spanish mackerel, ornithine decarboxylase, tea polyphenols

## Abstract

Putrescine is a low-molecular-weight organic compound that is widely found in pickled foods. Although the intake of biogenic amines is beneficial to humans, an excessive intake can cause discomfort. In this study, the ornithine decarboxylase gene (ODC) was involved in putrescine biosynthesis. After cloning, expression and functional verification, it was induced and expressed in *E. coli* BL21 (DE3). The relative molecular mass of the recombinant soluble ODC protein was 14.87 kDa. The function of ornithine decarboxylase was analyzed by determining the amino acid and putrescine content. The results show that the ODC protein could catalyze the decarboxylation of ornithine to putrescine. Then, the three-dimensional structure of the enzyme was used as a receptor for the virtual screening of inhibitors. The binding energy of tea polyphenol ligands to the receptor was the highest at −7.2 kcal mol^−1^. Therefore, tea polyphenols were added to marinated fish to monitor the changes in putrescine content and were found to significantly inhibit putrescine production (*p* < 0.05). This study lays the foundation for further research on the enzymatic properties of ODC and provides insight into an effective inhibitor for controlling the putrescine content in pickled fish.

## 1. Introduction

Amino acid decarboxylation results in the production of biologically active amines (BAs), which are commonly found in different foodstuffs, including dairy products [1], fish [2], fermented vegetables [3], alcoholic beverages [4], fresh meat and meat products [5]. Fish, fish derivatives and fermented goods are foods that may have high BA content. Additionally, BAs have been utilized as chemical markers of the standard of seafood [5]. BAs are primarily created by bacteria using substrate-specific decarboxylase enzymes to decarboxylate the relevant amino acids. Because they are frequently active at acidic pH levels, these enzymes may assist in retaining pH homeostasis or shorten the development time by detoxifying the extracellular medium [6]. Free amino acids, the existence of decarboxylase-positive microbes and favorable conditions for bacterial development are all factors that the procedure of fermentation of fish may give [7]. BAs are often produced in the process of pickling and fermenting fish. In a previous study, it was found that the Spanish mackerel can produce putrescine during the curing process. The strain in the Spanish mackerel was identified by 16s rRNA as *Enterobacter hormaechei*. It is usually believed that strains, rather than a particular species, are relevant to the production capabilities of BAs [8]. Enterobacteriaceae have shown great potential to produce polyamines, particularly putrescine, via the decarboxylation of ornithine [9]. The synthesis of putrescine in *E. coli* involves the decarboxylation of arginine by arginine decarboxylase to produce agmatine, which is then hydrolyzed by agmatine decarboxylation, or the hydrolysis of ornithine by ornithine decarboxylase [10].

BAs are commonly found in all types of seafood, influencing food quality, taste and nutrition, among other quality factors. However, BAs pose a serious threat to food safety. Recent years have shown an obvious increase in the frequency of amine poisoning. Thus, research on the production of fish BAs has increased, particularly with regard to the regulation mechanisms underlying their action. At present, studies have investigated the accumulation of BAs in the Spanish mackerel at home and abroad, although most have yet to explore them at the molecular level. In fact, a number of studies on the production of amines by the Spanish mackerel have only been reported by domestic scholars, with the mechanism underlying their action yet to be thoroughly discussed. From the aspects of food hygiene and security, the exploration and analysis of the formation mechanism of Spanish mackerel BAs mediated by spoilage microorganisms is not only effective for monitoring quality changes, but also necessary for the targeted suppression of food spoilage.

Common amine-producing strains include those from *Enterobacter*, *Enterococcus*, *Lactobacillus*, *Leuconostoc*, *Streptococcus*, *Oenococcus*, *Clostridium* and *Pseudomonas* [11]. Kim et al. studied the association between microbes and biogenic amine production in different fish products and found that Enterobacter aerogenes produce large amounts of histamine, cadaverine and putrescine during storage [12]. However, other Enterobacteriaceae strains mainly produce cadaverine and putrescine and produce less histamine. ODC is present in many bacteria in two different forms: a biosynthesis or constitutive form, and an able-to-breakdown or inducible form [13]. L-ornithine decarboxylase (ODC) converts ornithine to putrescine as the initial step in the production of polyamines. In eukaryotic systems, putrescine is one of the most strictly regulated enzymatic processes. Therefore, the ornithine decarboxylase gene (ODC) from the Spanish mackerel was cloned and expressed via polymerase chain reaction (PCR). In addition, the relationship between ODC expression levels and putrescine content was studied. A suitable inhibitor was found for the production of putrescine.

## 2. Materials and Methods

### 2.1. Materials

In an earlier study, a strain producing putrescine was obtained, which was identified by 16S rRNA as *Enterobacter hormaechei* (ID: CP036310.1). The purified strain was frozen at −80 °C until use.

### 2.2. Total DNA Extraction

The preserved strains were removed from a −80 °C ultra-low temperature freezer, and the plates were streaked on LB solid medium and inverted at 37 °C for 12–16 h. Single colonies were picked in 10 mL of LB liquid medium and incubated overnight at 37 °C under 160 rpm shaking. The bacterial solution was transferred to a 1.5 mL EP tube and centrifuged for 10 min to discard the supernatant. The cells were resuspended by adding 1 mL of TE buffer, and the cells were thoroughly vortexed. The bacterial solution was placed in a 1.5 mL EP tube and heated in a boiling water bath for 10 min. The supernatant was centrifuged and used as a template. The obtained DNA was dissolved in 50–100 μL of TE buffer and stored at −20 °C until use.

### 2.3. ODC Sequence and Cloning

Our team acquired the transcriptome data for *Enterobacter hormaechei* (NCBI accession: CP036310.1), and one ODC sequence was utilized as a template. The PCR amplification primers ODC-F and ODC-R (Table 1) were created using the Primer Premier 5.0 program. 

The PCR system (TaKaRa, Dalian, China) comprised the following reaction mixture (50 μL total volume): 10 μL of 5 × Prime STAR Buffer, 4 μL of 200 μM dNTPs, 1 μL of each primer (ODC-F and ODC-R), 4 μL of DNA template, 0.5 μL of Prime STAR HS DNA Polymerase and 50 μL of nuclease-free water in total. The reaction’s circumstances were as follows: thirty rounds in the denatured state at 98 °C for 10 s with annealing at 55 °C for 15 s and extending at 68 °C for 30 s.

A gel recovering kit (TaKaRa, Dalian) was utilized to obtain the PCR products. Using ampicillin resistance and blue-white screening, the extracted DNA was joined to the cloning vector pMD20-T and changed into capable cells of *Escherichia coli* JM109 (TaKaRa, Dalian). Bacterial PCR validation was carried out using the all-purpose primer M13 (Table 1), and eligible bacteria were sent to Takara Bio for sequencing.

### 2.4. ODC Bioinformatics Analysis

The ODC nucleic acid sequences obtained by sequencing were separately translated into amino acid sequences using the Edit Seq module of DNA Star software. The translated ODC protein’s amino acid sequence was submitted to the NCBI website for a sequence BLAST analysis, and the protein database was selected from the Swiss-Prot Database. The Prot-Param online program was used to examine the protein’s physicochemical properties, and SWISS MODEL was used to predict its three-dimensional structure.

### 2.5. ODC Prokaryotic Expression and Purification

According to the ODC sequence, a pair of primers (ODC-IF-F and ODC-IF-R) with restriction enzyme sites was constructed. (Table 1). The primers’ 5′ ends were modified to include the limitation enzyme sites of recognition for EcoR I and Hind III (TaKaRa, Dalian). The purified PCR products and pET-28a (+) vector were double-digested with EcoR I and Hind III restriction enzymes. To generate the recombinant plasmid pET-28a (+)/ODC, the products of the second digestion were attached for 15 min at 50 °C using 5 In-Fusion HD Enzyme Premix (TaKaRa, Dalian). The thermal shock approach was used to convert the modified plasmids into *E. coli* BL21 (DE3). For bacterial PCR verification, the regular primers T7 (Table 1) were employed. The identified strains were inoculated in medium, treated with isopropyl β-D-1-thiogalactopyranoside (IPTG) (1 mM), and cultured at 37 °C for 4 h. Electrophoresis on polyacrylamide gel with sodium dodecyl sulfate was performed in order to divide proteins on 12% gels. Western blotting was used to evaluate the expression of the target protein.

### 2.6. Evaluation of ODC Protein Function

*E. coli* BL21 (DE3) strains containing pET-28a (+) and pET-ODC plasmids were inoculated at a ratio of 1:100 (*v*/*v*) into 50 μg/mL of LB broth in 2 mL. After activating at 37 °C overnight, the activated bacterial solution was injected into 100 mL of LB liquid medium at a percentage of 1:50 (*v*/*v*). After culturing on a 37 °C shaker to an OD_595_ of 0.6, IPTG and L-ornithine hydrochloride were added to the shake flask to a final concentration of 1 mM and 2 μg/mL, respectively. The bacterial solution was spun around at 10,000 rpm for 10 min after being cultivated for 4 h. A similar volume of ethyl acetate that had been acidified with 0.1% glacial acetic acid was added after the bacterial fermentation supernatant was collected. The procedure of extraction involved shaking for 20 min, repeated three times. The organic phase obtained from the three extractions was collected and rotary evaporated to dryness. Then, 1 mL of methanol was integrated to it for reconstitution before storing at 4 °C for later use.

Enzymatic reactions were carried out in accordance with the literature, with slight variations, to evaluate the ODC protein’s role. In brief, the experimental and control groups were constructed as seen below (an overall amount of 2.5 mL): The untreated group received a 0.5 mL ornithine solution (10 mM), 50 g of deactivated protein boiled at a high temperature and 1 mL of 1 M K_2_CO_3_ (pH 8.0), whereas the experimental group received 1 mL of pyridoxal phosphate (PLP) solution (2 M). After being vortexed, tubes were placed in a water bath at 37 °C for 20 min. Within the test group, 50 g of pure protein was added, and the reaction was stopped with 1 mL of K_2_CO_3_ solution.

Then, taking 500 μL of the filtrate in a brown volumetric flask, 300 μL of concentrated NaHCO_3_ liquid and 200 μL of 2 mol/L NaOH solution were put into the buffer. Then, 1 mL of acetone and 10 mg/mL dansyl chloride solution were added, shaken well and placed in a 60 °C water bath to avoid light for 30 min. After taking it out and cooling it, 100 μL of ammonia water was added to interrupt the reaction, and it was let to stand for 20 min at room temperature. Then, the volumetric flask was placed in a 60 °C water bath for 15 min to remove the acetone. After taking it out and cooling, it was diluted to a 5 mL volumetric flask with acetonitrile. A 0.22 μm Millipore Corporation filter was used to filter the solution, and 10 μL of the filter fluid was used for the later use of HPLC testing. The following are the HPLC conditions: mobile phase A was water, mobile phase B was acetonitrile, and the flow rate was 1.0 mL/min on a Kromasil C18 column (250 mm, 4.6 mm, 5 m). The variation in elution conditions was as follows: 0–7 min, 45% to 35% phase A, 7–14 min, 35% to 30% phase A, 14–20 min, 30% phase A, 20–27 min, 30% to 10% phase A. The ODC protease reaction products were qualitatively analyzed via HPLC.

### 2.7. Screening of Targeted Inhibitors of Putrescine

According to the “National Food Safety Standard Food Additives Use Standards” and various other documents, some edible food additives were selected as candidate inhibitors for the screening of target inhibitors, as shown in Table 2. After washing the Spanish mackerel, the heads, tails and viscera were removed, and the samples were rinsed with tap water. As per the traditional process, the Spanish mackerel sample was mixed with 17% salt and additives (*w*/*w*) (tea polyphenols (0.3 g/kg), chlorogenic acid (0.045%), and sodium stearoyl lactylate (2.0 g/kg)). After the fish were marinated in salt for 6 h, saturated brine was added. After one day of curing, the samples were dried at 35 °C after controlling the moisture. The purpose was to remove a certain amount of water in order to obtain a certain salt content and inhibit the growth of microorganisms so that it could be preserved for a long time, which is a normal procedure for cured fish. The water content was about 20% of the fish. The samples were sealed and stored at 4 °C. Putrescine levels were measured every 4 days for 24 days to determine the effect of inhibitors on putrescine production and the control of fish spoilage.

### 2.8. Analysis of the Mechanism of ODC Protein Binding to the Ornithine Molecule

The ODC protein model was used as the receptor. With L-ornithine as the ligand, the 3D structure was from the PubChem database. Chem 3D software MM2 molecular mechanics was used to optimize the configuration of the compound. The tiny ligand molecule was then opened in PMV using AutodockTools1.5.6. After adding hydrogen and charge, the root of the ligand was detected. Autodock vina 1.1.2 was used for semi-flexible docking after searching for and specifying rotatable keys. Nine configurations were created as a result, with the best affinity conformation being chosen as the final docking conformation. Finally, the best ligand was obtained according to the binding energy score.

### 2.9. Statistical Analysis

The parameters were estimated using the IBM SPSS 13.0 software and an analysis of variance in one direction (ANOVA). The data are presented in the form of the mean standard deviation. Results with *p* values of 0.05 or lower were considered statistically significant.

## 3. Results

### 3.1. Cloning ODCs and Bioinformatics Evaluation

As seen in Figure 1A, the whole DNA of *E. hormaechei* was used to perform the PCR reaction of a gene component. The 481 bp-long coding region was cloned and sequenced. The open reading frame, however, was 390 bp. The protein’s relative molecular mass was 14.87 kDa, and its deduced amino acid sequence included 129 amino acids (molecular formula: C665H1051N187O186S7). The gene has the entry number WP 126486758.1 and was submitted to GenBank. The homology comparison’s outcomes showed that the ODC protein had a significant degree of commonality with other strains of ODC. (Figure 1B). In order to submit the ODC amino acid sequence, the protein database PBD ID: 1ord.1.A was used as a template in the online program SWISS-MODEL. The results showed that the homology between ODC and the ornithine decarboxylase gene was 49.19%, indicating that the model protein was ornithine decarboxylase.

### 3.2. ODC Expression and Protein Purification

The obtained target fragment ODC-PD was subjected to a ligation reaction with the linearized vector pET-28a (+)-EcoR I-Hind III, was then transferred into the host, namely *E. coli* BL21 (DE3) capable cells, and was sequenced using the vector. PCR was performed with primers T7/T7 terminator to screen for positive clones containing the recombinant plasmid. 

The successfully constructed expression strain pET28-ODC-BL21 was induced to express, and the control group comprised an empty plasmid containing the strain pET28-BL21. After IPTG was induced at a concentration of 1 mM and incubated at 37 °C for 4 h, The microbes were gathered, sonicated, centrifuged and loaded, followed by SDS-PAGE electrophoresis. A significant protein expression band at 14.9 kDa was observed in lanes 4 to 6 (Figure 1C).

The SDS gel electrophoresis showed that the size of the ODC protein was about 15 kDa. Therefore, the size of the expressed protein was consistent with its theoretical molecular weight. The Western blotting results (Figure 1D) show that the induced protein had a strong reaction band at the theoretical size, which was consistent with the SDS-PAGE results for the control. The expressed target protein was fused with the His tag protein, which also indicated that the constructed recombinant plasmid expressed the target protein correctly and was partially soluble in cells. In short, the expression of the target protein in *E. coli* provided favorable conditions for subsequent experiments.

### 3.3. ODC Protein Functional Verification

To explore the functional properties of the expressed protease, *E. coli* BL21 (DE3) cells containing the plasmids of pET-28a (+) (control group) and pET-ODC (experimental group) were inoculated into LB broth. After activation at 37 °C overnight, the cells were amplified and cultured. L-ornithine hydrochloride and the IPTG inducer were placed into the shaking flask at final concentrations of 1 mM and 2 g/mL after incubation to an OD595 of 0.6. After 4 h of incubation, the bacterial solution was centrifuged. The clarified fermentation broth was then collected. Equal amounts of ethyl acetate and 0.1% glacial acetic acid were added, shaken and extracted for 20 min. This procedure was repeated in triplicate. The organic phase obtained from the three extractions was collected and rotary evaporated. After the organic phase was evaporated to dryness, 1 mL of methanol was added for recombination. The amino acid content of the obtained samples was determined using a Hitachi L-8900 amino acid analyzer, and the results are shown in Figure 2.

The ornithine content in the control group was as high as 0.988 nmol/μL, whereas the content in the experimental group was only 0.312 nmol/μL. This was clearly the result of the action of ornithine decarboxylase. However, to further verify the decarboxylation function of ornithine decarboxylase, the putrescine content was also measured (Figure 2). It was further confirmed that ornithine decarboxylase plays a role in the conversion of ornithine into putrescine when the putrescine level of the test group treated with ornithine decarboxylase was considerably greater than that of the control group, reaching 58.72 mg/kg.

### 3.4. Screening of Targeted Inhibitors

The inhibitors were molecularly docked with ODC proteins for virtual screening. The affinity binding energies are shown in Table 3. Among them, tea polyphenols had the highest docking affinity with the protein, with a binding capacity of −7.2 kcal mol^−1^. Therefore, tea polyphenols were selected as the target inhibitors in vitro for the inhibition experiment.

During the curing of the Spanish mackerel, a control group and two experimental groups were set up. One of the control groups was the cured fish without any treatment, one of the experimental groups was treated with expressed ornithine decarboxylase, and the other was treated with tea polyphenol inhibitors. The putrescine content of the three groups was measured over 24 days (Figure 3). At the end of the whole curing cycle, the total putrescine content of the blank group increased by 0.511 mg/kg. Thus, the total putrescine content of Spanish mackerel gradually increased during the salting process, and the test group treated with ornithine decarboxylase had considerably higher putrescine content than that of the blank group. In addition, the putrescine content of the experimental group treated with tea polyphenol inhibitors was significantly reduced compared to that of the ornithine-decarboxylase-treated group, because the small molecules of tea polyphenol and ODC macromolecules were chelated (and had strong bond energy hydrogen bonds), thus preventing the formation of biogenic amines, indicating that tea polyphenol had some inhibitory effect on BAs.

To explore the mechanism of this phenomenon, the predicted ODC protein model was used as the receptor. L-ornithine and tea polyphenol molecules are ligands. The DS molecular docking software was used to analyze the interaction force between ODC proteins and small molecules, and the specific results of the interaction were obtained in the form of images. The binding affinity between the original ligand and the protein docking was −3.7 kcal mol^−1^. The theoretical binding mode is shown in Figure 4.

The theoretical combination of tea polyphenols is depicted in Figure 4. Among them, there were four bonds of hydrogen made with isoleucine at position 104, alanine at position 106 and glutamic acid at position 107, thereby forming a relatively stable complex with the ODC protein. Gerdt et al. studied the interaction force between receptor proteins and molecular binding and found that the contribution of hydrogen bonds is higher than that of other forces [14]. When Ding et al. used virtual screening to search for inhibitors of *p fluorescens*. receptors, they also believed that hydrogen bonds are the main force for the binding of small ligand molecules to proteins [15]. In summary, the ODC protein can bind tea polyphenol ligands through hydrogen bonding and hydrophobic interactions, and its main binding force is hydrogen bonding.

## 4. Discussion

BAs are nitrogen compounds with a minor molecular weight that are generated from amino acids. They are mostly generated in foods via the activity of decarboxylase [16]. It can also be generated through the transamination or amination of aldehydes and ketones [17]. According to their composition, they can be divided into two categories: monoamines and polyamines. Monoamines mainly include tryptamine, putrescine, cadaverine, histamine and tyramine, and polyamines mainly include spermine and spermidine [18]. A sufficient number of BAs is required for the regular physiological functioning of cells. Some are precursors of hormones, alkaloids, nucleic acids or protein synthesis [19]. They can promote growth and enhance metabolic vitality and immunity. Some BAs are important regulators of cell proliferation, differentiation, and gene expression. However, when ingested in excess, BAs can cause headaches, abdominal cramps, vomiting, and other adverse physiological reactions [20]. In previous studies, histamine was the most potentially dangerous alkaline food, with psychoactive effects and effects on the skin, as well as influencing blood vessels and causing gastrointestinal problems [21]. Although putrescine and cadaverine are not poisonous, they have been found to enhance histamine toxicity and negatively affect the sensory quality of foodstuffs [22,23]. Putrescine affects the organoleptic qualities of seafood and causes toxicological reactions, mostly by potentiating the poisonous effect of histamine by blocking intestinal histamine metabolizing enzymes such as diamine oxidase and histamine N-methyltransferase [24]. Additionally, a few BAs, including putrescine, cadaverine, spermidine and spermine, may combine with nitrogen dioxide to form volatile nitrosamines, which are substances that are thought to cause cancer [25].

Ornithine decarboxylase (ODC) generally exists in two forms in bacteria: one is biosynthetic or constitutive, whose enzyme is encoded by the speC gene, which is present in many putrescine-producing bacteria. The other is biodegradable or inducible, whose enzyme is encoded by the speF gene. These two genes are often denoted as ODC genes. Heimer et al. (1979) studied the activity of ornithine decarboxylase and found that ODC is the main method by which plants and vegetables synthesize putrescine, whereas in animal cells, putrescine is directly synthesized from ornithine through the action of ODC [26]. This may be due to the content of ornithine in plants being low, whereby the putrescine content in animal products is much higher by comparison. Generally, there are two reasons for the determination of putrescine in food: (i) putrescine is potentially toxic, and (ii) putrescine can serve as a sign of the nutritional value of food [27]. The most common sources of BA intoxication are histamine and tyramine [28]. Putrescine and cadaverine have no documented adverse health threat but may play an important role in food poisoning, as they can potentiate the toxicity of tyramine and histamine [29]. Furthermore, biogenic polyamines such as putrescine, spermine, spermidine and cadaverine may react with nitrite to form carcinogenic nitrosoamines [30]. In view of the potential relevance of biogenic amines for human health and food safety, it is of critical importance to monitor their levels in foodstuffs.

High BA concentrations can exist in foods such as fish and fish products, cheese, wine, sausages and fermented vegetables [31,32]. Fish is an important source of protein, vitamins and minerals. However, fish proteins are broken down rapidly when enzymes can be formed by bacterial growth, which generates biogenic amines, including histamine. Therefore, the quantification of amines in fish is considered a quality index and can be used as an important tool for sanitary surveillance [33].

In our previous work, we found that the Spanish mackerel produces a certain amount of putrescine during the curing process. This is because fish contain a high content of proteins and amino acids. Peptides and amino acids, the products of protein breakdown, provide precursors for the formation of BAs. Second, fish flesh is wealthy in nutrients and conducive to microbial development and procreation. Under the corresponding environmental conditions, it is easy to induce the accumulation of BAs during the processing of fish products [34]. Therefore, we screened for a putrescine-producing strain, identified as *Enterobacter hormaechei*. The target fragment of ornithine decarboxylase was successfully cloned using this bacterium as a DNA template and named ODC. The ODC sequence of different variants and the encoded amino acid sequence were similar in comparison, showing that ODC was highly conserved with other ornithine decarboxylase. ODC belongs to the ornithine decarboxylase family [35]. After informatics analysis, the expression was induced. The results show that ornithine decarboxylase is expressed in small amounts in the *E. coli* expression system, with a protein size of approximately 14.87 kDa. Subsequent in vitro experiments were performed to verify the functionality of the protein. The expressed protease has a certain decarboxylation function. Finally, the three-dimensional structure of the protein was used as the receptor for the virtual screening of putrescine inhibitors. Among the seven candidate food additives, tea polyphenols were found to have high binding energy to proteases. In recent studies, several potential mechanisms for reducing the production of BAs in salted fish have been reported. One strategy involves controlling the production of BAs in fermented foods by inhibiting the activity of microbial decarboxylase [36]. A study by Gupta et al. (2012) revealed that, after treating mice with the ODC inhibitor difluoromethyl-orntihine, the putrescine and spermidine content in cells was significantly reduced [37].

Another strategy is to reduce BA levels in fermented foods by using microorganisms that reduce BAs when added to fermented foods. Multiple microbial species interact during the complex biochemical process of fermenting fish, which is crucial for the buildup of BAs [38]. Amino acid decarboxylases, which are produced by some microorganisms, may be excreted; however, other microbes may also break down the BAs utilizing amine oxidases [39]. These microorganisms may be able to stop or lessen the buildup of BAs in food products, particularly fermented foods, and their potential role as BA preventers or reducers has drawn particular attention in recent years. Staphylococcus carnosus FS19 was shown to prevent the production of histamine in samples of fish sauce in researchand it noted that 18 and 21% salt seafood sauce samples, showed histamine production decreases of 15.1 and 13.8%, respectively [40]. Putrescine, histamine, tyramine and cadaverine significantly (*p* < 0.05) decreased when *Lactobacillus casei* CCDM 198 was used as the starting culture (up to 25% reduction in 48 h) [41]. Salted fish samples injected with *Bacillus polymyxis* D05-1 showed reductions in histamine and total BA levels of 34.0 and 30.0% [42,43]. *Pediococcus acidilactici* M28 could degrade all eight forms of BAs, and Staphylococcus carnosus M43 could degrade histamine and tyramine in the fermentation process [44].

The activity and expression of ODC influence the process of polyamine biosynthesis directly [45]. As a result, polyphenols from tea can decrease the activity of microbial decarboxylase while in the pickled fish fermentation process [46]. During the 24-day marinating period, the content of putrescine in Spanish mackerel rose from 0.372 to 0.883 mg/kg. When tea polyphenol inhibitors were added, the putrescine content was suppressed to an extent, resulting in a final putrescine content of only 1.390 mg/kg. Because BAs are generated by the enzymatic activity of microbes, this effect could be associated with the antibacterial capacity of tea polyphenols [47,48]. To explain this phenomenon, we conducted our analysis at the molecular level. First, the binding energy of decarboxylase receptors and tea polyphenols was significantly greater than that of the original ligand. This is most likely due to the four strong hydrogen bonds found between them. During the process, the inhibitor was found to significantly increase the proportion of binding to the receptor. Therefore, the activity of decarboxylase was inhibited, and the aim of inhibiting putrescine production was achieved.

## 5. Conclusions

According to our knowledge, this work is the first to successfully clone ODC from a Spanish mackerel, predict its physical, chemical and protein structure using bioinformatics analysis and confirm the function of ODC. As a consequence, there was a strong positive correlation between the putrescine content and the enzyme activity level of ODC. These findings serve as a foundation for further investigation into the biological properties of ODC genes, including the use of the structural properties of ODC to identify better putrescine inhibitors. Moreover, this study provides strong data supporting the improvement of the food safety of pickled aquatic products.

## Figures and Tables

**Figure 1 foods-12-02347-f001:**
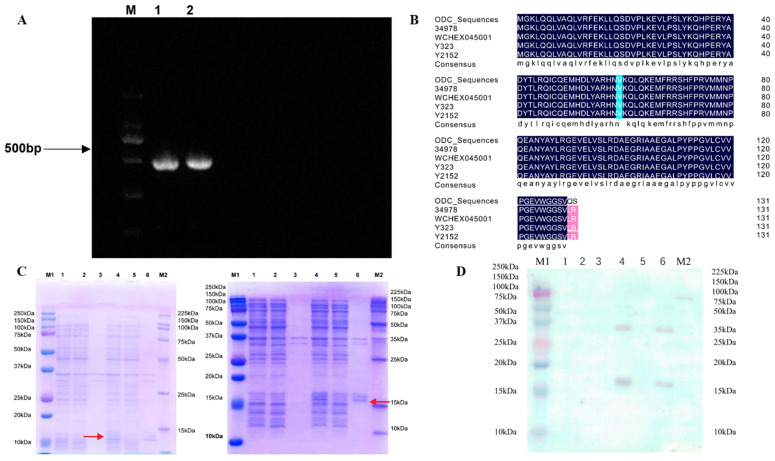
ODC protein (**A**). PCR amplification products of ODC fragments. (**B**). ODC amino acid sequence homology comparison with ODCs of other strains. (**C**). Expression of the target protein in *E. coli* BL21(DE3). SDS-PAGE of recombinant ODC protein expression. (**D**). Verification of the target protein via Western blotting.

**Figure 2 foods-12-02347-f002:**
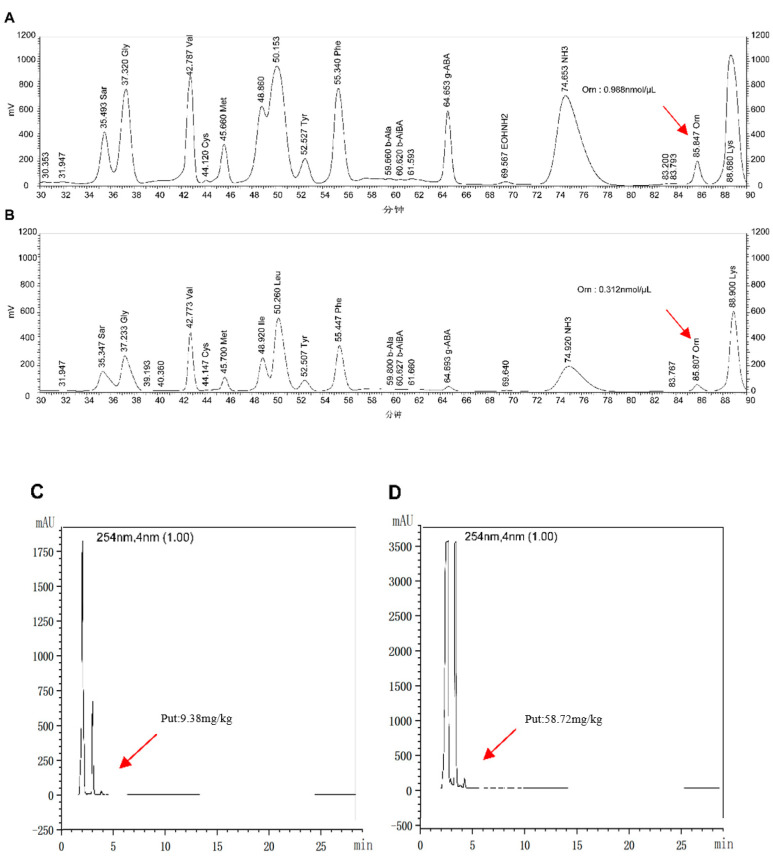
Functional verification of ODC protease. (**A**) *E. coli* BL21 (DE3) strains containing pET-28a (+) group for the determination of ornithine content; (**B**) *E. coli* BL21 (DE3) strains containing pET-ODC plasmids group for the determination of ornithine content; (**C**) *E. coli* BL21 (DE3) strains containing pET-28a (+) group for the determination of putrescine content; (**D**) *E. coli* BL21 (DE3) strains containing pET-ODC plasmids group for the determination of putrescine content.

**Figure 3 foods-12-02347-f003:**
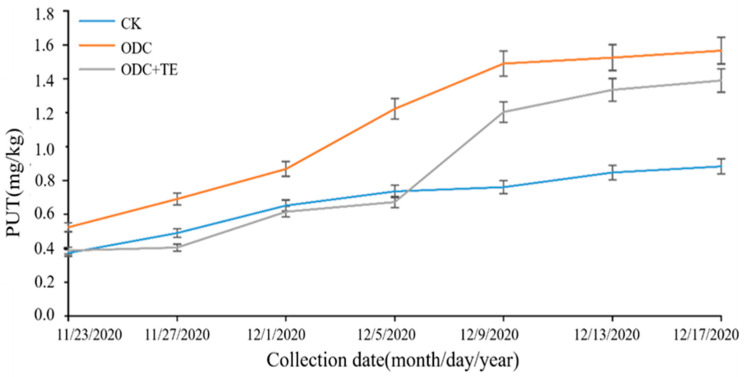
Changes in putrescine content in marinated Spanish mackerel.

**Figure 4 foods-12-02347-f004:**
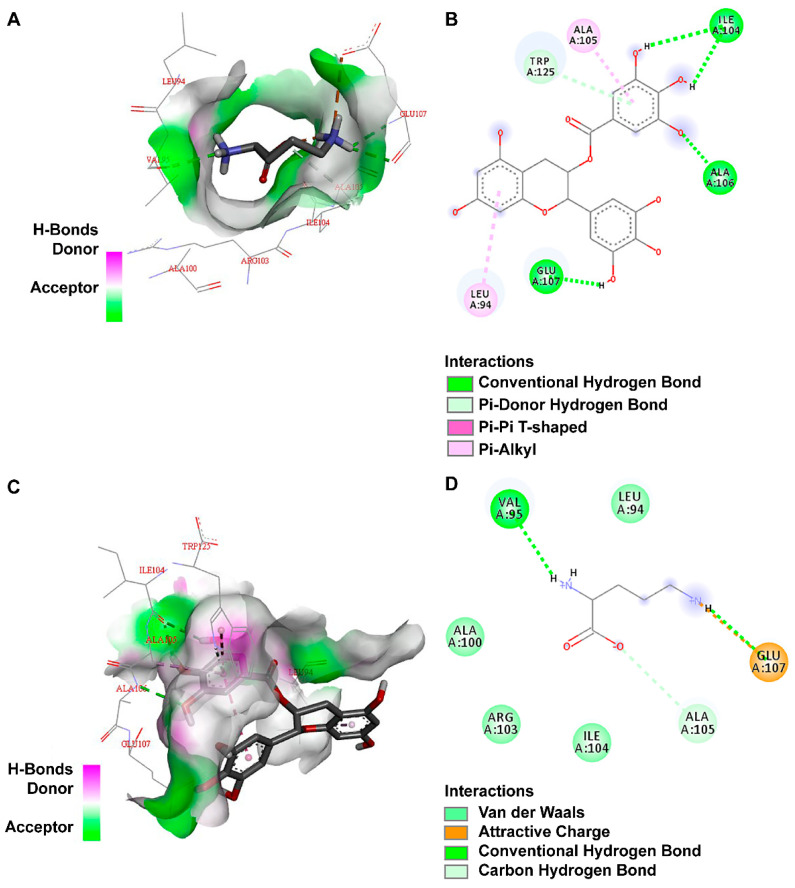
Molecular docking of the native ligands L-ornithine with the ODC protein models shown as 3D (**A**) and 2D (**B**) diagrams. Docked interactions of the tea polyphenols with the ODC protein model shown as 2D (**C**) and 3D (**D**) diagrams.

**Table 1 foods-12-02347-t001:** Sequences of the used primers.

Name of Primers	Sequences of Primers (5′ → 3′)
ODC-F	GCACGGAACCGCCCCAGACT
ODC-R	GCCACCATCCTCGCCAAC
ODC-IF-F	GACGGAGCTCGAATTATGGGTAAACTGCAGCAGCTG
ODC-IF-R	ACAATTCCCCTCTAGTTAAACAGAACCACCCCAAACTTC
M13-M4	GTTTTCCCAGTCACGAC
M13-RV	CAGGAAACAGCTATGAC
T7	TAATACGACTCACTATAGGG
T7 terminator	GCTAGTTATTGCTCAGCGG

**Table 2 foods-12-02347-t002:** Virtual screening inhibitor.

Name	CAS	Structural Formula
Sodium 1-carboxylatoethyl stearate	18200-72-1	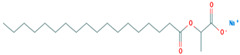
D-glucitol	50-70-4	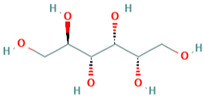
Tea polyphenol	84650-60-2	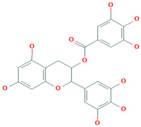
Poly(L-lysine) hydrobromide	25988-63-0	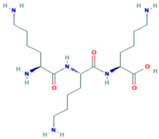
Gallic acid	149-91-7	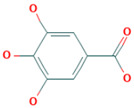
5-propyl-2-thiouracil	2954-52-1	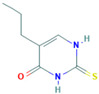
Chlorogenic acid	327-97-9	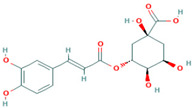

**Table 3 foods-12-02347-t003:** Inhibitor and protein binding ability score.

Name	CAS	Binding Energy (kcal mol^−1^)
Sodium 1-carboxylatoethyl Stearate	18200-72-1	−5.1
D-glucitol	50-70-4	−4.5
Tea polyphenol	84650-60-2	−7.2
Poly(L-lysine) hydrobromide	25988-63-0	−4.8
Gallic acid	149-91-7	−5.1
5-propyl-2-thiouracil	2954-52-1	−4.3
Chlorogenic acid	327-97-9	−6.7

## Data Availability

Data is contained within the article.

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
