# Peer review of "Characterization and Mechanism of Tea Polyphenols Inhibiting Biogenic Amine Accumulation in Marinated Spanish Mackerel"

_foods, 2023, doi:10.3390/foods12122347_

Round 1

Reviewer 1 Report

The authors have studied the ornithine synthesize decarboxlase gene (ODC) in order to produce putrescine. They further investigated the effect of tea polyphenols on the inhition of putrescine production in marinated fish. They demonstrated that ODC were able to catalyse ornithine to putrescine. Moreover, their study showed that tea polyphenols singificantly decreased the production of putrescine in marinated fish samples. 

Although the study is interesting in terms of food safety and quality control, there are some few points to be clarified and corrected in the manuscript. Mainly, the methodology section is not very clear. The authors must clarify the experimental groups and control group in the method section. The explanation on the methodology differs in some points too that must be corrected as well. For example, in the method section, the marinated samples stored for 28 days to be analysed, however, in the results section they mentioned 24 days. The methodology used in marination needs to be written in more detail. Why the authors dry the fish after marination? Is it the normal procedure for marination of fish or this step is carried out for sampling procedure? Moreover, in the results section, the authors written about the description of the groups as 'One group was treated with expressed ornithine decarboxylase and the other group was treated with tea polyphenol inhibitors'. However, the results showed that ODC+TE group has more putrescine value compared to control group. The authors should explain this section in more detail. 

The discussion section is very complicated and covers various repeated information. It also covers various irrelevant information relating to the study results. Therefore, the authors should revise this section.

There are some minor English errors which must be revised. 

Author Response

Response to Reviewer 1 Comments

Although the study is interesting in terms of food safety and quality control, there are some few points to be clarified and corrected in the manuscript. Mainly, the methodology section is not very clear. The authors must clarify the experimental groups and control group in the method section. The explanation on the methodology differs in some points too that must be corrected as well. For example, in the method section, the marinated samples stored for 28 days to be analysed, however, in the results section they mentioned 24 days. The methodology used in marination needs to be written in more detail. Why the authors dry the fish after marination? Is it the normal procedure for marination of fish or this step is carried out for sampling procedure? Moreover, in the results section, the authors written about the description of the groups as 'One group was treated with expressed ornithine decarboxylase and the other group was treated with tea polyphenol inhibitors'. However, the results showed that ODC+TE group has more putrescine value compared to control group. The authors should explain this section in more detail.

The discussion section is very complicated and covers various repeated information. It also covers various irrelevant information relating to the study results. Therefore, the authors should revise this section.

The authors corrected the manuscript as suggested. Hence, the paper is ready for publication after minor revision, as follows:

Point 1: Mainly, the methodology section is not very clear. The authors must clarify the experimental groups and control group in the method section. The explanation on the methodology differs in some points too that must be corrected as well.

Response 1: Thanks for the helpful comments. In the experimental methods section there were two different sets of experiments, one to assess ODC protein function, where the control group was E. coli strains with pET-28a(+) and the experimental group was E. coli strains containing the pET-ODC plasmid; the other set of experiments was to screen for targeting inhibitors, where control and experimental groups were established, where the control group was cured fish without any treatment and the experimental groups were cured fish treated with expressed ornithine decarboxylase treated cured fish and cured fish treated with teicoplanin inhibitor, respectively. The results are discussed in the latter part correspondingly in the revised manuscript.

Point 2: Why the authors dry the fish after marination? Is it the normal procedure for marination of fish or this step is carried out for sampling procedure?

Response 2: Thanks for the helpful comments. Fish should be stripped of a certain amount of water after curing to obtain a certain salt content to inhibit the growth of microorganisms so that it can be preserved for a long time, which is the normal procedure for cured fish [1].

[1] Zhang, J., Fang, Z., Cao, Y., Xu, Y., Hu, Y., Ye, X., & Yang, W. (2013). Effect of Different Drying Processes on the Protein Degradation and Sensory Quality of Layú: A Chinese Dry-Curing Grass Carp. DRYING TECHNOLOGY, 31(13-14), 1715-1722.

Point 3: For example, in the method section, the marinated samples stored for 28 days to be analysed, however, in the results section they mentioned 24 days. The methodology used in marination needs to be written in more detail.

Response 3: Thanks for the helpful comments. Cured samples should be stored for 24 days for analysis, and putrescine levels should be measured every 4 days for 24 days to determine the effect of inhibitors on putrescine production and control of fish spoilage, it has been revised in the revised manuscript.

Point 4: Moreover, in the results section, the authors written about the description of the groups as 'One group was treated with expressed ornithine decarboxylase and the other group was treated with tea polyphenol inhibitors. However, the results showed that ODC+TE group has more putrescine value compared to control group. The authors should explain this section in more detail.

Response 4: Thanks for the helpful comments. During salinization, the total putrescine content of Spanish mackerel gradually increased, and the putrescine content of the experimental group treated with ornithine decarboxylase was much higher than that of the blank group. And the putrescine content of the experimental group treated with tea polyphenol inhibitor was significantly reduced compared to the ornithine decarboxylase treated group, indicating that tea polyphenol had some inhibitory effect on BAs. Because the small molecules of tea polyphenols and ODC macromolecules were chelated (and had hydrogen bonds with strong bonding energy), which blocked the formation of biogenic amines.

Point 5: The discussion section is very complicated and covers various repeated information. It also covers various irrelevant information relating to the study results.

Response 5: Thanks for the helpful comments. The discussion section has been revised in the revised manuscript.

Reviewer 2 Report

As a researcher I am perhaps not the most qualified to evaluate applied genomics papers, but I very much appreciated the "application" spirit of this trial design. My compliments to the authors.
However, I have noticed many writing errors in the text, errors that I have pointed out in the pdf of my review that I attach to these notes.
I ask the authors for the patience to review the text where indicated by me.
For the rest, I believe that the article is valid for publication and I have no particular methodological remarks to make.

The article is written in good English, understandable, technicalities aside. I noticed some writing errors that I pointed out in the text of my review that I have attached to these notes.

Author Response

Response to Reviewer 2 Comments

Below are my feedbacks for the authors’ consideration:

The authors corrected the manuscript as suggested. Hence, the paper is ready for publication after minor revision, as follows:

Point: There are a number of writing errors in the text, which are noted in the attached PDF.

Response: Thanks for the helpful comments. The writing errors have been corrected in the revised manuscript.

Round 2

Reviewer 1 Report

The authors revised the manuscript properly. Therefore, it is suitable for publication.